# Dimensionality Reduction for Stationary Time Series via Stochastic Nonconvex Optimization

**Minshuo Chen**[1]    **Lin F. Yang**[2]    **Mengdi Wang**[2]    **Tuo Zhao**[1]
[1]Georgia Institute of Technology    [2]Princeton University
[1]{mchen393, tourzhao}@gatech.edu    [2]{lin.yang, mengdiw}@princeton.edu

## Abstract

Stochastic optimization naturally arises in machine learning. Efficient algorithms with provable guarantees, however, are still largely missing, when the objective function is nonconvex and the data points are dependent. This paper studies this fundamental challenge through a streaming PCA problem for stationary time series data. Specifically, our goal is to estimate the principle component of time series data with respect to the covariance matrix of the stationary distribution. Computationally, we propose a variant of Oja's algorithm combined with downsampling to control the bias of the stochastic gradient caused by the data dependency. Theoretically, we quantify the uncertainty of our proposed stochastic algorithm based on diffusion approximations. This allows us to prove the asymptotic rate of convergence and further implies near optimal asymptotic sample complexity. Numerical experiments are provided to support our analysis.

## 1  Introduction

Many machine learning problems can be formulated as a stochastic optimization problem in the following form,

$$\min_u \ \mathbb{E}_{Z \sim \mathcal{D}}[f(u, Z)] \quad \text{subject to } u \in \mathcal{U}, \tag{1.1}$$

where $f$ is a possibly nonconvex loss function, $Z$ denotes the random sample generated from some underlying distribution $\mathcal{D}$ (also known as statistical model), $u$ is the parameter of our interest, and $\mathcal{U}$ is a possibly nonconvex feasible set for imposing modeling constraints on $u$. For finite sample settings, we usually consider $n$ (possibly dependent) realizations of $Z$ denoted by $\{z_1, ..., z_n\}$, and the loss function in (1.1) is further reduced to an additive form, $\mathbb{E}[f(u, z)] = \frac{1}{n} \sum_{i=1}^{n} f(u, z_i)$. For continuously differentiable $f$, Robbins and Monro (1951) propose a simple iterative stochastic search algorithm for solving (1.1). Specifically, at the $k$-th iteration, we obtain $z_k$ sampled from $\mathcal{D}$ and take

$$u_{k+1} = \Pi_{\mathcal{U}}[u_k - \eta \nabla_u f(u_k, z_k)], \tag{1.2}$$

where $\eta$ is the step-size parameter (also known as the learning rate in machine learning literature), $\nabla_u f(u_k, z_k)$ is an unbiased stochastic gradient for approximating $\nabla_u \mathbb{E}_{Z \sim \mathcal{D}} f(u_k, Z)$, i.e.,

$$\mathbb{E}_{z_k} \nabla_u f(u_k, z_k) = \nabla_u \mathbb{E}_{Z \sim \mathcal{D}} f(u_k, Z),$$

and $\Pi_{\mathcal{U}}$ is a projection operator onto the feasible set $\mathcal{U}$. This seminal work is the foundation of the research on stochastic optimization, and has a tremendous impact on the machine learning community.

The theoretical properties of such a stochastic gradient descent (SGD) algorithm have been well studied for decades, when both $f$ and $\mathcal{U}$ are convex. For example, Sacks (1958); Bottou (1998); Chung (2004); Shalev-Shwartz et al. (2011) show that under various regularity conditions, SGD converges to a global optimum as $k \to \infty$ at different rates. Such a line of research for convex and smooth objective function $f$ is fruitful and has been generalized to nonsmooth optimization (Duchi et al., 2012b; Shamir and Zhang, 2013; Dang and Lan, 2015; Reddi et al., 2016).

When $f$ is nonconvex, which appears more often in machine learning problems, however, the theoretical studies on SGD are very limited. The main reason behind is that the optimization landscape of nonconvex problems can be much more complicated than those of convex ones. Thus, conventional optimization research usually focuses on proving that SGD converges to first order optimal stationary solutions (Nemirovski et al., 2009). More recently, some results in machine learning literature show that SGD actually converges to second order optimal stationary solutions, when the nonconvex optimization problem satisfies the so-called "strict saddle property" (Ge et al., 2015; Lee et al., 2017). More precisely, when the objective has negative curvatures at all saddle points, SGD can find a way to escape from these saddle points. A number of nonconvex optimization problems in machine learning and signal processing have been shown to satisfy this property, including principal component analysis (PCA), multiview learning, phase retrieval, matrix factorization, matrix sensing, matrix completion, complete dictionary learning, independent component analysis, and deep linear neural networks (Srebro and Jaakkola, 2004; Sun et al., 2015; Ge et al., 2015; Sun et al., 2016; Li et al., 2016; Ge et al., 2016; Chen et al., 2017).

These results further motivate many followup works. For example, Allen-Zhu (2017) improves the iteration complexity of SGD from $\widetilde{O}(\epsilon^{-4})$ in Ge et al. (2015) to $\widetilde{O}(\epsilon^{-3.25})$ for general unconstrained functions, where $\epsilon$ is a pre-specifed optimization accuracy; Jain et al. (2016); Allen-Zhu and Li (2016) show that the iteration complexity of SGD for solving the eigenvalue problem is $\widetilde{O}(\epsilon^{-1})$. Despite of these progresses, we still lack systematic approaches for analyzing the algorithmic behavior of SGD. Moreover, these results focusing on the convergence properties, however, cannot precisely capture the uncertainty of SGD algorithms (e.g., how to escape from saddle points), which makes the theoretical analysis less intuitive.

Besides nonconvexity, data dependency is another important challenge arising in stochastic optimization for machine learning, since the samples $z_k$'s are often collected with a temporal pattern. For many applications (e.g., time series analysis), this may involve certain dependency. Taking generalized vector autoregressive (GVAR) data as an example, our observed $z_{k+1} \in \mathbb{R}^m$ is generated by $z_{k+1}^i | z_k \sim p(a_i^\top z_k)$, where $a_i$'s are unknown coefficient vectors, $z_{k+1}^i$ is the $i$-th component of $z_{k+1}$, $p(\cdot)$ denotes the density of the exponential family, and $a_i^\top z_k$ is the natural parameter. There is only limited literature on convex stochastic optimization for dependent data. For example, Duchi et al. (2012a) investigate convex stochastic optimization algorithms for ergodic underlying data generating processes; Homem-de Mello (2008) investigates convex stochastic optimization algorithms for dependent but identically distributed data. For nonconvex optimization problems in machine learning, however, how to address such dependency is still quite open.

This paper proposes to attack stochastic nonconvex optimization problems for dependent data by investigating a simple but fundamental problem in machine learning — Streaming PCA for stationary time series. PCA has been well known as a powerful tool to reduce the dimensionality, and well applied to data visualization and representation learning. Specifically, we solve the following nonconvex problem,

$$U^* \in \underset{U \in \mathbb{R}^{m \times r}}{\operatorname{argmax}} \operatorname{Trace}(U^\top \Sigma U) \quad \text{subject to } U^\top U = I_r \tag{1.3}$$

where $\Sigma$ is the covariance matrix of our interest. This is also known as an eigenvalue problem. The column span of the optimal solution $U^*$ equals the subspace spanned by the eigenvectors corresponding to the first $r$ largest eigenvalues of $\Sigma$. Existing literature usually assumes that at the $k$-th iteration, we observe a random vector $z_k$ independently sampled from some distribution $\mathcal{D}$ with $\mathbb{E}[z_k] = 0$ and $\mathbb{E}[z_k z_k^\top] = \Sigma$. Our setting, however, assumes that $z_k$ is sampled from some time series with a stationary distribution $\pi$ satisfying $\mathbb{E}_\pi[z_k] = 0$ and $\mathbb{E}_\pi[z_k z_k^\top] = \Sigma$. There are two key computational challenges in such a streaming PCA problem:

• For time series, it is difficult to get unbiased estimators of the covariance matrix of the stationary distribution because of the data dependency. Taking GVAR as an example, the marginal distribution of $z_k$ is different from the stationary distribution. As a result, the stochastic gradient at the $k$-th iteration is biased, i.e., $\mathbb{E}[z_k z_k^\top U_k | U_k] \neq \Sigma U_k$;

• The optimization problem in (1.3) is nonconvex, and its solution space is rotational-invariant. Given any orthogonal matrix $Q \in \mathbb{R}^{r \times r}$ and any feasible solution $U$, the product $UQ$ is also a feasible solution and gives the same column span as $U$. When $r > 1$, this fact leads to the degeneracy in the optimization landscape such that equivalent saddle points and optima are non-isolated. The algorithmic behavior under such degeneracy is still quite open for SGD.

To address the first challenge, we propose a variant of Oja's algorithm to handle data dependency. Specifically, inspired by Duchi et al. (2012a), we use downsampling to generate weakly dependent samples. Theoretically, we show that the downsampled data point yields a sequence of stochastic approximations of the covariance matrix of the stationary distribution with controllable small bias. Moreover, the block size for downsampling only logarithmically depends on the optimization accuracy, which is nearly constant (see more details in Sections 2 and 3).

To attack nonconvexity and the degeneracy of the solution space, we establish new convergence analysis based on principle angle between $U_k$ and the eigenspace of $\Sigma$. By applying diffusion approximations, we show that the solution trajectory weakly converges to the solution of a stochastic differential equation (SDE), which enables us to quantify the uncertainty of the proposed algorithm (see more details in Sections 3 and 5). Investigating the analytical solution of the SDE allows us to characterize the algorithmic behavior of SGD in three different scenarios: escaping from saddle points, traversing between stationary points, and converging to global optima. We prove that the stochastic algorithm asymptotically converges and achieves nearly optimal asymptotic sample complexity.

There are several closely related works. Chen et al. (2017) study the streaming PCA problem for $r = 1$ also based on diffusion approximations. However, $r = 1$ makes problem (1.3) admit an isolated optimal solution, unique up to sign change. For $r > 1$, the global optima are nonisolated due to the rotational invariance property. Thus, the analysis is more involved and challenging. Moreover, Jain et al. (2016); Allen-Zhu and Li (2016) provide nonasymptotic analysis for the Oja's algorithm for streaming PCA. Their techniques are quite different from ours. Their nonasymptotic results, though more rigorous in describing discrete algorithms, lack intuition and can only be applied to the Oja's algorithm with no data dependency. In contrast, our analysis handles data dependency and provides detailed explanation to the asymptotic algorithmic behavior.

**Notations:** Given a vector $v = (v_1, \ldots, v_m)^\top \in \mathbb{R}^m$, we define the Euclidean norm $\|v\|_2^2 = v^\top v$. Given a matrix $A \in \mathbb{R}^{m \times n}$, we define the spectral norm $\|A\|_2$ as the largest singular value of $A$ and the Frobenius norm $\|A\|_F^2 = \mathrm{Trace}(AA^\top)$. We also define $\sigma_r(A)$ as the $r$-th largest singular value of $A$. For a diagonal matrix $\Theta \in \mathbb{R}^{m \times m}$, we define $\sin \Theta = \mathrm{diag}\,(\sin(\Theta_{11}), \ldots, \sin(\Theta_{mm}))$ and $\cos \Theta = \mathrm{diag}\,(\cos(\Theta_{11}), \ldots, \cos(\Theta_{mm}))$. We denote the canonical basis of $\mathbb{R}^m$ by $e_i$ for $i = 1, \ldots, m$ with the $i$-th element being 1, and the canonical basis of $\mathbb{R}^r$ by $e_j'$ for $j = 1, \ldots, r$.

## 2 Downsampled Oja's Algorithm

We first explain how to construct a nearly unbiased covariance estimator for the stationary distribution, which is crucial for our proposed algorithm. Before proceed, we briefly review geometric ergodicity for time series, which characterizes the mixing time of a Markov chain.

**Definition 2.1** (Geometric Ergodicity and Total Variation Distance)**.** A Markov chain with state space $S$ and stationary distribution $\pi$ on $(S, \mathcal{F})$ with $\mathcal{F}$ being a $\sigma$-algebra on $S$, is geometrically ergodic, if it is positive recurrent and there exists an absolute constant $\rho \in (0, 1)$ such that the total variation distance satisfies

$$\mathcal{D}_{\mathrm{TV}}\,(p^n(x, \cdot), \pi(\cdot)) = \sup_{A \in \mathcal{F}} |p^n(x, A) - \pi(A)| = O\,(\rho^n) \quad \text{for all } x \in S,$$

where $p^n(\cdot, \cdot)$ is the $n$-step transition kernel[1].

Note that $\rho$ is independent of $n$ and only depends on the underlying transition kernel of the Markov chain. The geometric ergodicity is equivalent to saying that the chain is $\beta$-mixing with an exponentially decaying coefficient (Bradley et al., 2005).

As aforementioned, one key challenge of solving the streaming PCA problem for time series is that it is difficult to get unbiased estimators of the covariance matrix $\Sigma$ of the stationary distribution. However, when the time series is geometrically ergodic, the transition probability $p^h(z_k, z_{k+h})$ converges exponentially fast to the stationary distribution. This allows us to construct a nearly unbiased estimator of $\Sigma$ as shown in the next lemma.

**Lemma 2.2.** Let $\{z_k\}_{k=1}^{\infty}$ be a geometrically ergodic Markov chain with parameter $\rho$, and assume $z_k$ is Sub-Gaussian. Given a pre-specified accuracy $\tau$, there exists $h = O\left(\kappa_\rho \log \frac{1}{\tau}\right)$ such that

$$\mathbb{E}\left[(z_{2h+k} - z_{h+k})(z_{2h+k} - z_{h+k})^\top / 2 \Big| z_k \right] = \Sigma + E\Sigma$$

with $\|E\|_2 \leq \tau$, where $\kappa_\rho$ is a constant depending on $\rho$ and $\Sigma$ is the covariance matrix of $z_k$ under the stationary distribution.

Lemma 2.2 shows that as $h$ increases, the bias decreases to zero. This suggests that we can use the downsampling method to reduce the bias of the stochastic gradient. Specifically, we divide the data points into blocks of length $2h$, i.e., $\boxed{z_1, z_2, \ldots, z_{2h}}$, $\boxed{z_{2h+1}, \ldots, z_{4h}}$, $\ldots$, $\boxed{z_{2(b-1)h+1}, \ldots, z_{2bh}}$. For the $s$-th block, we use data points $z_{(2s-1)h}$ and $z_{2sh}$ to approximate $\Sigma$ by $X_s = \frac{1}{2}(z_{2sh} - z_{(2s-1)h})(z_{2sh} - z_{(2s-1)h})^\top$. Later we will show that the block size $h$ only logarithmically depends on the optimization accuracy. Thus, the downsampling is affordable. Moreover, if the stationary distribution has zero mean, we only need the block size to be $h$ and $X_s = z_{sh}z_{sh}^\top$.

Many time series models in machine learning are geometrically ergodic. We discuss a few examples.

**Example 2.3.** The vector autoregressive (VAR) model follows the update $z_{k+1} = Az_k + \epsilon_k$, where $\epsilon_k$'s are i.i.d. Sub-Gaussian random vectors with $\mathbb{E}[\epsilon_k] = 0$ and $\mathbb{E}[\epsilon_k \epsilon_k^\top] = \Gamma$, and $A$ is the coefficient matrix. When $\rho = \|A\|_2 < 1$, the model is stationary and geometrically ergodic (Tjøstheim, 1990). Moreover, the mean of its stationary distribution is 0.

**Example 2.4.** Recall that GVAR model follows $z_{k+1}^i | z_k \sim p(a_i^\top z_k)$, where $z_{k+1}^i$'s are independent conditioning on $z_k$. The density function is $p(x|\theta) = h(x) \exp\left(T(x)\theta - B(\theta)\right)$, where $T(x)$ is a statistic, and $B(\theta)$ is the log partition function. GVAR is stationary and geometrically ergodic under certain regularity conditions (Hall et al., 2016).

As an illustrative example, we show that for Gaussian VAR with $\rho = \|A\|_2 < 1$ and $\Gamma = I$, the bias of the covariance estimator can be controlled by choosing $h = O\left(\frac{1}{1-\rho} \log \frac{1}{\tau}\right)$. The covariance matrix of the stationary distribution is $\Sigma = \sum_{i=0}^{\infty} A^i (A^\top)^i$. One can check

$$\mathbb{E}\left[z_{h+k} z_{h+k}^\top | z_k\right] - \Sigma = A^h z_k z_k^\top (A^\top)^h + \sum_{i=h}^{\infty} A^i (A^\top)^i.$$

Here the spectrum of $A$ acts as the geometrically decaying factor for both terms on the right hand side, since both terms are of the order $O(\rho^{2h})$. As a result, the bias of $\mathbb{E}\left[z_{h+k} z_{h+k}^\top | z_k\right]$ decays to zero exponentially fast. We pick $h = O\left(\frac{1}{1-\rho} \log \frac{1}{\tau}\right)$, and obtain $\mathbb{E}\left[z_{k+h} z_{k+h}^\top | z_k\right] = \Sigma + E\Sigma$ with $\|E\|_2 \leq \tau$.

We then propose a variant of Oja's algorithm combined with our downsampling technique as summarized in Algorithm 1. For simplicity, we assume the stationary distribution has mean zero. The projection $\Pi_{\text{Orth}}(U)$ denotes the orthogonalization operator that performs on columns of $U$. Specifically, for $U \in \mathbb{R}^{m \times r}$, $\Pi_{\text{Orth}}(U)$ returns a matrix $U' \in \mathbb{R}^{m \times r}$ that has orthonormal columns. Typical examples of such operators include Gram-Schmidt method and Householder transformation. The step,

$$U_{s+1} = \Pi_{\text{Orth}}(U_s + \eta X_s U_s),$$

is essentially the original Oja's update. Our variant manipulates on data points by downsampling such that $X_s$ is nearly unbiased. We emphasize that $s$ denotes the number of iterations, and $k$ denotes the number of samples.

---

**Algorithm 1** Downsampled Oja's Algorithm

**Input:** data points $z_k$, block size $h$, step size $\eta$
Initialize $U_1$ with orthonormal columns.
Set $s \leftarrow 1$
**repeat**
    Take sample $z_{sh}$, and set $X_s \leftarrow z_{sh} z_{sh}^\top$
    $U_{s+1} \leftarrow \Pi_{\text{Orth}}(U_s + \eta X_s U_s)$
    $s \leftarrow s + 1$
**until** Convergence
**Output:** $U_s$

---

## 3 Theory

We exploit diffusion approximations to characterize the convergence of downsampled SGD in 3 stages. Specifically, we use an ODE (Theorem 3.4) to analyze the global convergence and SDEs (Theorems 3.5 and 3.8) to capture the local dynamics around saddle points and global optima. By the weak

convergence of the discrete algorithm trajectory to the ODE and SDE, we show that downsampled SGD achieves an nearly optimal asymptotic sample complexity (Corollary 3.10). Before proceed, we impose some model assumptions on the problem.

**Assumption 3.1** . There exists an eigengap in the covariance matrix $\Sigma$ of the stationary distribution, i.e., $\lambda_1 \geq \cdots \geq \lambda_r > \lambda_{r+1} \geq \cdots \geq \lambda_m > 0$, where $\lambda_i$ is the $i$-th eigenvalue of $\Sigma$.

**Assumption 3.2** . Data points $\{z_k\}_{k \geq 1}$ are generated from a geometrically ergodic time series with parameter $\rho$, and the stationary distribution has mean zero. Each $z_k$ is Sub-Gaussian, and the block size is chosen as $h = O\left(\kappa_\rho \log(1/\eta)\right)$ for downsampling.

The eigengap in Assumption 3.1 implies that the optimal solution is identifiable. Specifically, the optimal solution $U^*$ is unique up to rotation. The positive definite assumption on $\Sigma$ is for theoretical simplicity. Assumption 3.2 implies that each $z_k$ has bounded moments of any order.

We also briefly explain the optimization landscape of streaming PCA problems as follows. Specifically, we consider the eigenvalue decomposition $\Sigma = R \Lambda R^\top$ with $\Lambda = \text{diag}(\lambda_1, \lambda_2, \dots, \lambda_m)$. Recall that $e_i$ is the canonical basis of $\mathbb{R}^m$. We distinguish stationary points $U$ of streaming PCA problems:

• $U$ is a global optimum, if the column span of $R^\top U$ equals the subspace spanned by $\{e_1, \dots, e_r\}$;

• $U$ is a saddle point or a global minima, if the column span of $R^\top U$ equals the subspace spanned by $\{e_{a_1}, \dots, e_{a_r}\}$, where $\mathcal{A}_r = \{a_1, \dots, a_r\} \subset \{1, \dots, m\}$ and $\mathcal{A}_r \neq \{1, \dots, r\}$.

For convenience, if the column span of $R^\top U$ coincides with $\{e_{a_1}, \dots, e_{a_r}\}$, we say that $U$ is a stationary point corresponding to the set $\mathcal{A}_r = \{a_1, \dots, a_r\}$.

To handle the rotational invariance of the solution space, we use principle angle to characterize the distance between the column spans of $U^*$ and $U_s$. The notation is as follows. Given two matrices $U \in \mathbb{R}^{m \times r_1}$ and $V \in \mathbb{R}^{m \times r_2}$ with orthonormal columns, where $1 \leq r_1 \leq r_2 \leq m$, the principle angle between these two matrices is, $\Theta(U, V) = \text{diag}\left(\arccos\left(\sigma_1(U^\top V)\right), \dots, \arccos\left(\sigma_{r_1}(U^\top V)\right)\right).$

We show the consequence of using principle angle as follows. Specifically, any optimal solution $U^*$ satisfies $\left\| \sin \Theta(R_r, U^*) \right\|_F^2 = \left\| \cos \Theta(\overline{R}_r, U^*) \right\|_F^2 = 0$, where $R_r$ denotes the first $r$ columns of $R$, and $\overline{R}_r$ denotes the last $m - r$ columns of $R$. This essentially implies that the column span of $U^*$ is orthogonal to that of $\overline{R}_r$. Thus, to prove the convergence of SGD, we only need to show $\left\| \cos \Theta(\overline{R}_r, U_s) \right\|_F^2 \to 0$. By the rotational invariance of principle angle, we obtain $\Theta\left(\overline{R}_r, U_s\right) = \Theta\left(R^\top \overline{R}_r, R^\top U_s\right) = \Theta\left(\overline{E}_r, R^\top U_s\right)$, where $\overline{E}_r = [e_{r+1}, \dots, e_m]$. For notational simplicity, we denote $\overline{U}_s = R^\top U_s$. Then the convergence of the algorithm is equivalent to $\left\| \cos \Theta\left(\overline{E}_r, \overline{U}_s\right) \right\|_F^2 \to 0$. We need such an orthogonal transformation, because $\left\| \cos \Theta\left(\overline{E}_r, \overline{U}_s\right) \right\|_F^2$ can be expressed as $\left\| \cos \Theta\left(\overline{E}_r, \overline{U}_s\right) \right\|_F^2 = \sum_{i=r+1}^m \left\| e_i^\top \overline{U}_s \right\|_2^2 = \sum_{i=r+1}^m \gamma_{i,s}^2$ with $\gamma_{i,s}^2 = \left\| e_i^\top \overline{U}_s \right\|_2^2$.

### 3.1 Global Convergence by ODE

Since the sequence $\{z_{sh}, \overline{U}_s\}_{s=1}^\infty$ forms a discrete Markov process, we can apply diffusion approximations to establish global convergence of SGD. Specifically, by a continuous time interpolation, we construct continuous time processes $U^\eta(t)$ and $X^\eta(t)$ such that $U^\eta(t) = U_{\lfloor t/\eta \rfloor + 1}$ and $X^\eta(t) = X_{\lfloor t/\eta \rfloor + 1}$. The subscript $\lfloor t/\eta \rfloor + 1$ denotes the number of iterations, and the superscript $\eta$ highlights the dependence on $\eta$. We denote $\overline{U}^\eta(t) = R^\top U^\eta(t)$ and $\overline{X}^\eta(t) = R^\top X^\eta(t) R$. The continuous time version of $\gamma_{i,s}^2$ is written as $\gamma_{i,\eta}^2(t) = \|e_i^\top \overline{U}^\eta(t)\|_2^2$. It is difficult to directly characterize the global convergence of $\gamma_{i,\eta}^2(t)$. Thus, we introduce an upper bound of $\gamma_{i,\eta}^2(t)$ as follows.

**Lemma 3.3.** Let $E_r = [e_1, \dots, e_r] \in \mathbb{R}^{m \times r}$. Suppose $\overline{U}^\eta(t)$ has orthonormal columns and $E_r^\top \overline{U}^\eta(t)$ is invertible. We have

$$\widetilde{\gamma}_{i,\eta}^2(t) = \left\| e_i^\top \overline{U}^\eta(t) \left( E_r^\top \overline{U}^\eta(t) \right)^{-1} \right\|_2^2 \geq \gamma_{i,\eta}^2(t). \tag{3.1}$$

The detailed proof is provided in Appendix B.1. We show $\widetilde{\gamma}_{i,\eta}^2(t)$ converges in the following theorem.

**Theorem 3.4.** As $\eta \to 0$, the process $\widetilde{\gamma}_{i,\eta}^2(t)$ weakly converges to the solution of the ODE

$$d\widetilde{\gamma}_i^2 = b_i \widetilde{\gamma}_i^2 dt \quad \text{with} \quad b_i \leq 2(\lambda_i - \lambda_r), \tag{3.2}$$

where $\widetilde{\gamma}_i^2(0) = \left\| e_i^\top \overline{U}(0) \left( E_r^\top \overline{U}(0) \right)^{-1} \right\|_2^2$, and $\overline{U}(0)$ has orthonormal columns.

The detailed proof is provided in Appendix B.2. The analytical solution to (3.2) is $\widetilde{\gamma}_i^2(t) = \widetilde{\gamma}_i^2(0)e^{b_i t}$ with $b_i \leq 2(\lambda_{r+1} - \lambda_r) < 0$ for any $i \in \{r+1, \ldots, m\}$. Note that we need $E_r^\top \overline{U}(0)$ to be invertible to derive the upper bound (3.1). Under this condition, $\widetilde{\gamma}_i^2(t)$ converges to zero. However, when $E_r^\top \overline{U}(0)$ is not invertible, the algorithm starts at a saddle point, and (3.2) no longer applies. As can be seen, the ODE characterization is insufficient to capture the local dynamics (e.g., around saddle points or global optima) of the algorithm.

### 3.2 Local Dynamics by SDE

The deterministic ODE characterizes the average behavior of the solution trajectory. To capture the uncertainty of the local algorithmic behavior, we need to rescale the influence of the noise to bring the randomness back, which leads us to a stochastic differential equation (SDE) approximation.

• **Stage 1: Escape from Saddle Points** Recall that $\Lambda = \mathrm{diag}(\lambda_1, \ldots, \lambda_m)$ collects all the eigenvalues of $\Sigma$. We consider the eigenvalue decomposition $\overline{U}^\top(0)\Lambda\overline{U}(0) = Q^\top \widetilde{\Lambda} Q$, where $Q \in \mathbb{R}^{r \times r}$ is orthogonal and $\widetilde{\Lambda} = \mathrm{diag}(\widetilde{\lambda}_1, \ldots, \widetilde{\lambda}_r)$. Again, by a continuous time interpolation, we denote $\zeta_{ij,\eta}(t) = \eta^{-1/2} e_j'^\top Q[\overline{U}^\eta(t)]^\top e_i$, where $e_j'$ is the canonical basis in $\mathbb{R}^r$. Then we decompose the principle angle $\gamma_{i,\eta}^2(t)$ as $\gamma_{i,\eta}^2(t) = \eta \sum_{j=1}^r \zeta_{ij,\eta}^2(t)$. Recall that $\overline{U}(0)$ is a saddle point, if the column span of $\overline{U}(0)$ equals the subspace spanned by $\{e_{a_1}, \ldots, e_{a_r}\}$ with $\mathcal{A}_r = \{a_1, \ldots, a_r\} \neq \{1, \ldots, r\}$. Therefore, if the algorithm starts around a saddle point, there exists some $i \in \{1, \ldots, r\}$ such that $\gamma_{i,\eta}^2(0) \approx 0$ and $\gamma_{a,\eta}^2(0) \approx 1$ for $a \in \mathcal{A}_r$. The asymptotic behavior of $\gamma_{i,\eta}^2(t)$ around a saddle point is captured in the following theorem.

**Theorem 3.5.** Suppose $\overline{U}(0)$ is initialized around a saddle point corresponding to $\mathcal{A}_r$. As $\eta \to 0$, conditioning on the event $\left\{\gamma_{i,\eta}^2(t) = O(\eta) \text{ for some } i \in \{1, \ldots, r\}\right\}$, $\zeta_{ij,\eta}(t)$ weakly converges to the solution of the following stochastic differential equation

$$d\zeta_{ij} = K_{ij}\zeta_{ij}dt + G_{ij}dB_t \quad \text{with} \quad K_{ij} \in [\lambda_i - \lambda_1, \lambda_i - \lambda_{a_r}] \text{ and } G_{ij}^2 < \infty, \qquad (3.3)$$

where $B_t$ is a standard Brownian motion, and $a_r$ is the largest element in $\mathcal{A}_r$.

The detailed proof is provided in Appendix B.3. We remark that the event $\gamma_{i,\eta}^2(t) = O(\eta)$ is only a technical assumption. This does not cause any issue, since when $\eta^{-1}\gamma_{i,\eta}^2(t)$ is large, the algorithm has escaped from the saddle point. Note that (3.3) admits the analytical solution

$$\zeta_{ij}(t) = \zeta_{ij}(0)e^{K_{ij}t} + G_{ij}\int_0^t e^{-K_{ij}(s-t)}dB(s), \qquad (3.4)$$

which is known as an O-U process. We give the following implications on different values of $K_{ij}$:

**(a)**. When $K_{ij} > 0$, rewrite (3.4) as $\zeta_{ij}(t) = \left[\zeta_{ij}(0) + G_{ij}\int_0^t e^{-K_{ij}s}dB(s)\right]e^{K_{ij}t}$. The exponential term $e^{K_{ij}t}$ is dominant and increases to positive infinity as $t \to \infty$. While the remaining part on the right hand side is a process with mean $\zeta_{ij}(0)$ and variance bounded by $G_{ij}^2/(2K_{ij})$. Hence, $e^{K_{ij}t}$ acts as a driving force to increase $\zeta_{ij}(t)$ exponentially fast so that $\zeta_{ij}(t)$ quickly gets away from 0;

**(b)**. When $K_{ij} < 0$, the mean of $\zeta_{ij}(t)$ is $\zeta_{ij}(0)e^{K_{ij}t}$. The initial condition restricts $\zeta_{ij}(0)$ to be small. Thus as $t$ increases, the mean of $\zeta_{ij}(t)$ converges to zero. Thus, the drift term vanishes quickly. The variance of $\zeta_{ij}(t)$ is bounded by $-G_{ij}^2/(2K_{ij})$. Hence, $\zeta_{ij}(t)$ roughly oscillates around 0;

**(c)**. When $K_{ij} = 0$, the drift term is approximately zero, meaning that $\zeta_{ij}(t)$ also oscillates around 0.

We provide an example showing how the algorithm escapes from a saddle point. Suppose that the algorithm starts at the saddle point corresponding to $\mathcal{A}_r = \{1, \ldots, q-1, q+1, \ldots, r, p\}$. Consider the principle angle $\gamma_{q,\eta}^2(t)$. By implication **(a)**, we have $K_{qr} = \lambda_q - \lambda_p > 0$. Hence $\zeta_{qr,\eta}(t)$ increases quickly away from zero. Thus, $\gamma_{q,\eta}^2(t) \geq \eta\zeta_{qr,\eta}^2(t)$ also increases quickly, which drives the algorithm away from the saddle point. Meanwhile, by **(b)** and **(c)**, $\gamma_{i,\eta}^2(t)$ stays at 1 for $i < q$ because of the vanishing drift. The algorithm tends to escape from the saddle point through reducing $\gamma_{p,\eta}^2(t)$, since this yields the largest eigengap, $\lambda_q - \lambda_p$. When we have $q = r$ and $p = r+1$, the eigengap is minimal. Thus, it is the worst situation for the algorithm to escape from a saddle point. We give the following proposition characterizing the time for the algorithm to escape from a saddle point.

**Proposition 3.6.** Suppose that the algorithm starts around the saddle point corresponding to $\mathcal{A}_r = \{1, \ldots, r-1, r+1\}$. Given a pre-specified $\nu$ and $\delta = O(\eta^{\frac{1}{2}})$ for a sufficiently small $\eta$, we need

$$T_1 \asymp \frac{1}{\lambda_r - \lambda_{r+1}} \log(K+1) \quad \text{with} \quad K = \frac{2(\lambda_r - \lambda_{r+1})\eta^{-1}\delta^2}{\left[\Phi^{-1}\left(\frac{1-\nu/2}{2}\right)\right]^2 G_{rr}^2},$$

such that $\mathbb{P}\left(\gamma_{r,\eta}^2(T_1) \geq \delta^2\right) \geq 1 - \nu$, where $\Phi$ is the CDF of the standard Gaussian distribution.

The detailed proof is provided in Appendix B.4. This implies that, asymptotically, we need

$$S_1 \asymp \frac{T_1}{\eta} \asymp \frac{\log(K+1)}{\eta(\lambda_r - \lambda_{r+1})}$$

iterations to escape from a saddle point, and the algorithm enters the second stage.

• **Stage 2: Traverse between Stationary Points**  After the algorithm escapes from the saddle point, the gradient is dominant, and the noise is negligible. Thus, the algorithm behaves like an almost deterministic traverse between stationary points, which can be viewed as a two-step discretization of the ODE with an error of the order $O(\eta)$ (Griffiths and Higham, 2010). Hence, we focus on $\gamma_{i,\eta}^2(t)$ to characterize the algorithmic behavior in this stage. Recall that we assume $\mathcal{A}_r = \{1, \ldots, r-1, r+1\}$. When the algorithm escapes from the saddle point, we have $\gamma_{r,\eta}^2(T_1) \geq \delta^2$, which implies $\sum_{i=r+1}^m \gamma_{i,\eta}^2(t) \leq 1 - \delta^2$. The following proposition assumes that the algorithm starts at this initial condition.

**Proposition 3.7.** Restarting the counter of time, for a sufficiently small $\eta$ and $\delta = O(\eta^{\frac{1}{2}})$. We need

$$T_2 \asymp \frac{1}{\lambda_r - \lambda_{r+1}} \log \frac{1}{\delta^2}$$

such that $\mathbb{P}\left(\sum_{i=r+1}^m \gamma_{i,\eta}^2(t) \leq \delta^2\right) \geq \frac{3}{4}$.

The detailed proof is provided in Appendix B.5. This implies that, asymptotically, we need

$$S_2 \asymp \frac{T_2}{\eta} \asymp \frac{1}{\eta(\lambda_r - \lambda_{r+1})} \log \frac{1}{\delta^2}$$

iterations to reach the neighborhood of the global optima.

• **Stage 3: Converge to Global Optima**  Similar to stage 1, we focus on $\zeta_{ij,\eta}(t)$ to characterize the dynamics of the algorithm around the global optima using an SDE approximation.

**Theorem 3.8.** Suppose $\overline{U}(0)$ is initialized around the global optima with $\sum_{i=r+1}^m \gamma_{i,\eta}^2(0) = O(\eta)$. Then as $\eta \to 0$, for $i = r+1, \ldots, m$ and $j = 1, \ldots, r$, $\zeta_{ij,\eta}(t)$ weakly converges to the solution of the following SDE

$$d\zeta_{ij} = K_{ij}\zeta_{ij}dt + G_{ij}dB_t \quad \text{with} \quad K_{ij} \in [\lambda_i - \lambda_1, \lambda_i - \lambda_r] \text{ and } G_{ij}^2 < \infty, \quad (3.5)$$

where $B_t$ is a standard Brownian motion.

The detailed proof is provided in Appendix B.6. The analytical solution of (3.5) is

$$\zeta_{ij}(t) = \zeta_{ij}(0)e^{K_{ij}t} + G_{ij}\int_0^t e^{-K_{ij}(s-t)}dB(s).$$

We then establish the following proposition.

**Proposition 3.9.** For a sufficiently small $\epsilon$ and $\eta$, $\delta = O(\eta^{\frac{1}{2}})$, restarting the counter of time, we need

$$T_3 \asymp \frac{1}{\lambda_r - \lambda_{r+1}} \log K' \quad \text{with} \quad K' = \frac{8(\lambda_r - \lambda_{r+1})\delta^2}{(\lambda_r - \lambda_{r+1})\epsilon - 4\eta r G_m},$$

such that we have $\mathbb{P}\left(\sum_{i=r+1}^m \gamma_{i,\eta}^2(T_3) \leq \epsilon\right) \geq \frac{3}{4}$, where $G_m = \max_{1 \leq j \leq r} \sum_{i=r+1}^m G_{ij}^2$.

The detailed proof is provided in Appendix B.7. The subscript $m$ in $G_m$ highlights its dependence on the dimension $m$. Proposition 3.9 implies that, asymptotically, we need

$$S_3 \asymp \frac{T_3}{\eta} \asymp \frac{\log K'}{\eta(\lambda_r - \lambda_{r+1})}$$

iterations to converge to an $\epsilon$-optimal solution in the third stage. Combining all the results in three stages, we know that after $T_1 + T_2 + T_3$ time, the algorithm converges to an $\epsilon$-optimal solution asymptotically. This further leads us to a more refined result in the following corollary.

**Corollary 3.10.** For a sufficiently small $\epsilon$, we choose
$$\eta \asymp \frac{(\lambda_r - \lambda_{r+1})\epsilon}{5rG_m}.$$
Suppose we start the algorithm near a saddle point, then we need $T = T_1 + T_2 + T_3$ such that
$$\mathbb{P}\left( \left\| \cos \Theta \left( \overline{E}_r, \overline{U}^\eta(T) \right) \right\|_{\mathrm{F}}^2 \leq \epsilon \right) \geq \frac{3}{4}.$$

The detailed proof is provided in Appendix B.8. Recall that we choose the block size $h$ of downsampling to be $h = O\left( \kappa_\rho \log \frac{1}{\eta} \right)$. Thus, the asymptotic sample complexity satisfies
$$N \asymp \frac{Th}{\eta} \asymp \frac{rG_m}{\epsilon(\lambda_r - \lambda_{r+1})^2} \log^2 \frac{rG_m}{\epsilon(\lambda_r - \lambda_{r+1})}.$$
From the perspective of statistical recovery, the obtained estimator $\widehat{U}$ enjoys a near-optimal asymptotic rate of convergence $\left\| \cos \Theta(\widehat{U}, U^*) \right\|_{\mathrm{F}}^2 \asymp \frac{rG_m \log N}{(\lambda_r - \lambda_{r+1})^2 N / \kappa_\rho}$, where $N$ is the number of data points.

## 4 Numerical Experiments

We demonstrate the effectiveness of our proposed algorithm using both simulated and real datasets.

• **Simulated Data** We first verify our analysis of streaming PCA problems for time series using a simulated dataset. We choose a Gaussian VAR model with dimension $m = 16$. The random vector $\epsilon_k$'s are independently sampled from $N(0, S)$, where
$$S = \mathrm{diag}(1, 1, 1, 1, 1, 1, 1, 1, 1, 1, 1, 1, 3, 3, 3).$$
We choose the coefficient matrix $A = V^\top D V$, where $V \in \mathbb{R}^{16 \times 16}$ is an orthogonal matrix that we randomly generate, and $D = 0.1 D_0$ is a diagonal matrix satisfying
$$D_0 = \mathrm{diag}(0.68, 0.68, 0.69, 0.70, 0.70, 0.70, 0.72, 0.72, 0.72, 0.72, 0.72, 0.72, 0.80, 0.80, 0.85, 0.90).$$
By solving the discrete Lyapunov equation $\Sigma = A\Sigma A^\top + S$, we calculate the covariance matrix of the stationary distribution, which satisfies $\Sigma = U^\top \Lambda U$, where $U \in \mathbb{R}^{16 \times 16}$ is orthogonal and
$$\Lambda = \mathrm{diag}(3.0175, 3.0170, 3.0160, 1.0077, 1.0070, 1.0061, 1.0058, 1.0052,$$
$$1.0052, 1.0052, 1.0052, 1.0051, 1.0049, 1.0049, 1.0047, 1.0047).$$
We aim to find the leading principle components of $\Sigma$ corresponding to the first 3 largest eigenvalues. Thus, the eigengap is $\lambda_3 - \lambda_4 = 2.0083$. We initialize the solution at the saddle point whose column span is the subspace spanned by the eigenvectors corresponding to 3.0175, 3.0170 and 1.0077. The step size is $\eta = 3 \times 10^{-5}$, and the algorithm runs with $8 \times 10^5$ total samples. The trajectories of the principle angle over 20 independent simulations with block size $h = 4$ are shown in Figure 1a. We can clearly distinguish three different stages. Figure 1c and 1d illustrate that entries of principle angles, $\zeta_{33}$ in stage 1 and $\zeta_{42}$ in stage 3, are Ornstein-Uhlenbeck processes. Specifically, the estimated distributions of $\zeta_{33}$ and $\zeta_{42}$ over 100 simulations follow Gaussian distributions. We can check that the variance of $\zeta_{33}$ increases in stage 1 as iteration increases, while the variance of $\zeta_{42}$ in stage 3 approaches a fixed value. All these simulated results are consistent with our theoretical analysis.

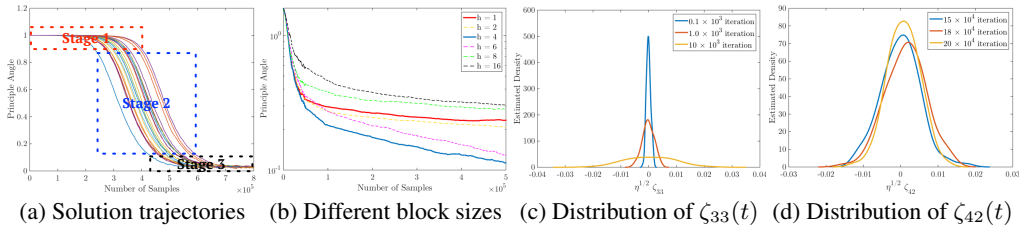

(a) Solution trajectories (b) Different block sizes (c) Distribution of $\zeta_{33}(t)$ (d) Distribution of $\zeta_{42}(t)$

Figure 1: Illustrations of various algorithmic behaviors in simulated examples: (a) presents three stages of the algorithm; (b) compares the performance of different block sizes; (c) and (d) demonstrate the Ornstein-Uhlenbeck processes of $\zeta_{33}$ in stage 1 and $\zeta_{42}$ in stage 3.

We further compare the performance of different block sizes of downsampling with step size annealing. We keep using Gaussian VAR model with $D = 0.9 D_0$ and
$$S = \mathrm{diag}(1.45, 1.45, 1.45, 1.45, 1.45, 1.45, 1.45, 1.45, 1.45, 1.45, 1.45, 1.45, 1.45, 1.455, 1.455, 1.455).$$

The eigengap is $\lambda_3 - \lambda_4 = 0.005$. We run the algorithm with $5 \times 10^5$ samples and the chosen step sizes vary according to the number of samples $k$. Specifically, we set the step size $\eta = \eta_0 \times \frac{h}{4000}$ if $k < 2 \times 10^4$, $\eta = \eta_0 \times \frac{h}{8000}$ if $k \in [2 \times 10^4, 5 \times 10^4)$, $\eta = \eta_0 \times \frac{h}{48000}$ if $k \in [5 \times 10^4, 10 \times 10^4)$, and $\eta = \eta_0 \times \frac{h}{120000}$ if $k \geq 10 \times 10^4$. We choose $\eta_0$ in $\{0.125, 0.25, 0.5, 1, 2\}$ and report the final principle angles achieved by different block sizes $h$ in Table 1. Figure 1b presents the averaged principle angle over 5 simulations with $\eta_0 = 0.5$. As can be seen, choosing $h = 4$ yields the best performance. Specifically, the performance becomes better as $h$ increases from 1 to around 4. However, the performance becomes worse, when $h = 16$ because of the lack of iterations.

Table 1: The final principle angles achieved by different block sizes with varying $\eta_0$.

|  | $\eta_0 = 0.125$ | $\eta_0 = 0.25$ | $\eta_0 = 0.5$ | $\eta_0 = 1$ | $\eta_0 = 2$ |
|---|---|---|---|---|---|
| $h = 1$ | 0.7775 | 0.3595 | **0.2320** | 0.2449 | 0.3773 |
| $h = 2$ | 0.7792 | 0.3569 | **0.2080** | 0.2477 | 0.2290 |
| $h = 4$ | 0.7892 | 0.3745 | **0.1130** | 0.3513 | 0.4730 |
| $h = 6$ | 0.7542 | 0.3655 | **0.1287** | 0.3317 | 0.3983 |
| $h = 8$ | 0.7982 | 0.3933 | **0.2828** | 0.3820 | 0.4102 |
| $h = 16$ | 0.7783 | 0.4324 | **0.3038** | 0.5647 | 0.6526 |

• **Real Data** We adopt the Air Quality dataset (De Vito et al., 2008), which contains 9358 instances of hourly averaged concentrations of totally 9 different gases in a heavily polluted area. We remove measurements with missing data. We aim to estimate the first 2 principle components of the series. We randomly initialize the algorithm, and choose the block size of downsampling to be 1, 3, 5, 10, and 60. Figure 2 shows that the projection of each data point onto the leading and the second principle components. We also present the result of projecting data points onto the eigenspace of sample covariance matrix indicated by Batch in Figure 2. All the projections have been rotated such that the leading principle component is parallel to the horizontal axis. As can be seen, when $h = 1$, the projection yields some distortion in the circled area. When $h = 3$ and $h = 5$, the projection results are quite similar to the Batch result. As $h$ increases, however, the projection displays obvious distortion again compared to the Batch result. The concentrations of gases are naturally time dependent. Thus, we deduce that the distortion for $h = 1$ comes from the data dependency, while for the case $h = 60$, the distortion comes from the lack of updates. This phenomenon coincides with our simulated data experiments.

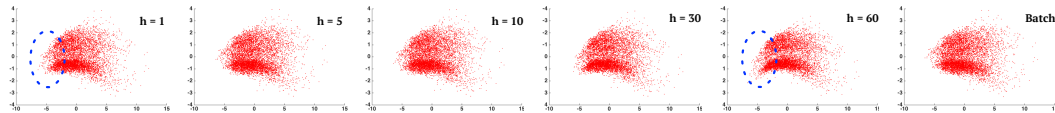

Figure 2: Projections of air quality data onto the leading and the second principle components with different block sizes of downsampling. We highlight the distortions for $h = 1$ and $h = 60$.

## 5    Discussions

We remark that our analysis characterizes how our proposed algorithm escapes from the saddle point. This is not analyzed in the related work, Allen-Zhu and Li (2016), since they use random initialization. Note that our analysis also applies to random initialization, and directly starts with the second stage.

Our analysis is inspired by diffusion approximations in existing applied probability literature (Glynn, 1990; Freidlin and Wentzell, 1998; Kushner and Yin, 2003; Ethier and Kurtz, 2009), which target to capture the uncertainty of stochastic algorithms for general optimization problems. Without explicitly specifying the problem structures, these analyses usually cannot lead to concrete convergence guarantees. In contrast, we dig into the optimization landscape of the streaming PCA problem. This eventually allows us to precisely characterize the algorithmic dynamics and provide concrete convergence guarantees, which further lead to a deeper understanding of the uncertainty in nonconvex stochastic optimization.

The block size $h$ of downsampled Oja's algorithm is based on the mixing property of the time series. We believe estimating the mixing coefficient is an interesting problem. The procedure in Hsu et al. (2015) estimates the mixing time of Markov chains, which may possibly be adapted to our time series setting. Moreover, estimating the covariance matrix of the stationary distribution is also interesting but challenging. We leave them for future investigation.

## Footnotes

[1]The formal definitions of positive recurrent and transition kernel can be found in Durrett (2010) Chapter 6. In short, a positive recurrent Markov chain visits each state in a finite time almost surely, and transition kernel is a generalization of transition probability to continuous state spaces.

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
