[Supplementary Material]

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

# A    Detailed Proofs in Section 2

## A.1    Proof of Lemma 2.2

*Proof.* We first assume the stationary distribution has zero mean and denote the covariance matrix as $\Sigma$. The total variation distance of $p^h(z, \cdot)$ and $\pi(\cdot)$ is equivalent to

$$\mathcal{D}_{\mathrm{TV}}(p^h(z, \cdot), \pi(\cdot)) = \frac{1}{2} \int \left| p^h(z, x) - \pi(x) \right| dx.$$

Then we try to find the conditional expectation,

$$\mathbb{E}\left[ z_{k+h} z_{k+h}^\top \big| z_k \right] = \int x x^\top p^h(z_k, x) dx$$

$$= \int x x^\top (\pi(x) + p^h(z_k, x) - \pi(x)) dx$$

$$= \int x x^\top \pi(x) dx + \int x x^\top (p^h(z_k, x) - \pi(x)) dx$$

$$= \Sigma + \int x x^\top (p^h(z_k, x) - \pi(x)) dx.$$

We bound the second term by the following,

$$\left\| \int x x^\top (p^h(z_k, x) - \pi(x)) dx \right\|_2 \leq \int \|x\|_2^2 \left| p^h(z_k, x) - \pi(x) \right| dx$$

$$\leq \int_{\|x\|_2^2 \leq t} \|x\|_2^2 \left| p^h(z_k, x) - \pi(x) \right| dx + \int_{\|x\|_2 > t} \|x\|_2^2 \left| p^h(z_k, x) - \pi(x) \right| dx$$

$$\leq C_1 t \rho^h + \int_{\|x\|_2^2 > t} \|x\|_2^2 \left| p^h(z_k, x) - \pi(x) \right| dx$$

$$\leq C_1 t \rho^h + \int_{\|x\|_2^2 > t} \|x\|_2^2 p^h(z_k, x) dx + \int_{\|x\|_2^2 > t} \|x\|_2 \pi(x) dx$$

$$\leq C_1 t \rho^h + \int_t^\infty \mathbb{P}_{p^h}(\|x\|_2^2 > s) ds + \int_t^\infty \mathbb{P}_\pi(\|x\|_2^2 > s) ds.$$

By our assumption, $x$ is a Sub-Gaussian random vector, then $\mathbb{P}_{p^h}(\|x\|_2 > t) \leq C_2 \exp(-C_3 t^2)$ and $\mathbb{P}_\pi(\|x\|_2 > t) \leq C_2' \exp(-C_3' t^2)$. The integration is bounded by

$$\int_{\sqrt{t}}^\infty \exp(-s^2) ds = \int_0^\infty \exp(-(s + \sqrt{t})^2) ds \leq \exp(-t) \int_0^\infty \exp(-2s\sqrt{t}) ds = \frac{1}{\sqrt{t}} \exp(-t).$$

Thus, we have $\left\| \int x x^\top (p^h(z_k, x) - \pi(x)) dx \right\|_2 \leq C_1 t \rho^h + C_2 \frac{1}{\sqrt{t}} e^{-C_3 t}$. Optimize over $t$ and neglect the exponential term, we pick $t = O\left( \rho^{-\frac{2h}{3}} \right)$ to reach $\left\| \int x x^\top (p^h(z_k, x) - \pi(x)) dx \right\|_2 \leq O(\rho^{h/3})$. Therefore, we have $\mathbb{E}\left[ z_{k+h} z_{k+h}^\top \big| z_k \right] = \Sigma + E\Sigma$ with $\|E\|_2 = O(\rho^{h/3})$, which implies that if we pick $h = O\left( \frac{1}{1-\rho} \log \frac{1}{\tau} \right)$, then we have $\|E\|_2 \leq \tau$.

For the general case, i.e., the stationary distribution has nonzero mean $\mu$, we proceed with double conditioning. Specifically, we calculate

$$\mathbb{E}\left[ (z_{k+2h} - z_{k+h})(z_{k+2h} - z_{k+h})^\top \big| z_k \right] = \mathbb{E}\left[ \mathbb{E}\left[ (z_{k+2h} - z_{k+h})(z_{k+2h} - z_{k+h})^\top \big| z_{k+h}, z_k \right] \big| z_k \right].$$

Then by the Markov property, the inner expectation is equal to $\mathbb{E}\left[ (z_{k+2h} - z_{k+h})(z_{k+2h} - z_{k+h})^\top \big| z_{k+h} \right]$. By a similar reasoning to the zero mean case, we first calculate the conditional expectation

$$\mathbb{E}\left[ (z_{k+2h} - z_{k+h})(z_{k+2h} - z_{k+h})^\top \big| z_{k+h} \right] = \Sigma + \mu \mu^\top - \mu z_{k+h}^\top - z_{k+h} \mu^\top + z_{k+h} z_{k+h}^\top + W,$$

where the remainder $W$ satisfies $\|W\|_2 = O(\rho^{h/3})$. Then taking expectation conditioning on $z_k$, we can derive

$$\mathbb{E}\left[ \frac{1}{2}(z_{k+2h} - z_{k+h})(z_{k+2h} - z_{k+h})^\top \big| z_k \right] = \Sigma + E\Sigma \text{ with } \|E\|_2 = O\left( e^{h\kappa_\rho} \right).$$

The calculation is a repetition of the zero mean case with the extra mean term $\mu$.    $\square$

# B  Detailed Proofs in Section 3

## B.1  Proof of Lemma 3.3

*Proof.* We omit the time indicator $t$. Since $\overline{U}$ and $E_r$ has orthonormal columns, we have $\left\|E_r^\top \overline{U}\right\|_2 \le 1$. Thus

$$\gamma_i(t) = \left\|e_i^\top \overline{U}\right\|_2 \le \left\|e_i^\top \overline{U}(E_r^\top \overline{U})^{-1}(E_r^\top \overline{U})\right\|_2 \le \left\|e_i^\top \overline{U}(E_r^\top \overline{U})^{-1}\right\|_2 \left\|E_r^\top \overline{U}\right\|_2 \le \left\|e_i^\top \overline{U}(E_r^\top \overline{U})^{-1}\right\|_2 = \widetilde{\gamma}_i(t)$$

$\square$

## B.2  Proof of Theorem 3.4

*Proof.* Compute the infinitesimal increments of $\widetilde{\gamma}_i^2(t)$, which is defined to be

$$\Delta\widetilde{\gamma}_{i,s}^2(t) = \widetilde{\gamma}_{i,s+1}^2 - \widetilde{\gamma}_{i,s}^2.$$

The sequence $\{z_{sh}, \overline{U}_s\}$ forms a Markov chain. By *Corollary 4.2 of chapter 7.4* of Ethier and Kurtz (2009), once

$$b_i = \lim_{\eta \to 0} \mathbb{E}\left[\frac{\Delta\widetilde{\gamma}_i^2(t)}{\eta}\bigg|\overline{U}_s, z_{sh}\right] < \infty,$$

$$\sigma_i^2 = \lim_{\eta \to 0} \mathbb{E}\left[\frac{[\Delta\widetilde{\gamma}_i^2(t)]^2}{\eta}\bigg|\overline{U}_s, z_{sh}\right] = 0,$$

the sequence $\widetilde{\gamma}_{i,s}^2(t)$ weakly converges to the solution of the following ODE,

$$d\widetilde{\gamma}_i^2 = b_i\widetilde{\gamma}_i^2 dt.$$

Hence, we must find the mean and variance of $\Delta\widetilde{\gamma}_{i,s}^2(t)$. For simplicity, we omit the subscript $s$.

$$\Delta\widetilde{\gamma}_{i,s}^2 = e_i^\top(\overline{U}+\Delta\overline{U})(E_r^\top(\overline{U}+\Delta\overline{U}))^{-1}(E_r^\top(\overline{U}+\Delta\overline{U}))^{-\top}(\overline{U}+\Delta\overline{U})^\top e_i - e_i^\top \overline{U}(E_r^\top\overline{U})^{-1}(E_r^\top\overline{U})^{-\top}\overline{U}^\top e_i$$

$$= 2e_i^\top\Delta\overline{U}(E_r^\top\overline{U})^{-\top}(E_r^\top\overline{U})^{-1}\overline{U}^\top e_i - 2e_i^\top\overline{U}(E_r^\top\overline{U})^{-1}(E_r^\top\overline{U})^{-\top}(E_r^\top\Delta\overline{U})^\top(E_r^\top\overline{U})^{-\top}\overline{U}^\top e_i + O(\|\Delta\overline{U}\|_2^2)$$

$$= 2\eta e_i^\top\overline{U}(E_r^\top\overline{U})^{-1}(E_r^\top\overline{U})^{-\top}\overline{U}^\top \overline{X}e_i - 2\eta e_i^\top\overline{U}(E_r^\top\overline{U})^{-1}(E_r^\top\overline{U})^{-\top}(E_r^\top \overline{X}\overline{U})^\top(E_r^\top\overline{U})^{-\top}\overline{U}^\top e_i + O(\eta^2\|\overline{X}\|_2^2)$$

where $\Delta\overline{U} = \eta(I - \overline{U}\,\overline{U}^\top)\overline{X}\,\overline{U} + O(\eta^2\overline{X}^2)$. We have used the fact that

$$(E_r^\top(\overline{U}+\Delta\overline{U}))^{-1} = ((E_r^\top\overline{U})(I + (E_r^\top\overline{U})^{-1}(E_r^\top\Delta\overline{U})))^{-1}$$

$$= (I - (E_r^\top\overline{U})^{-1}(E_r^\top\Delta\overline{U}) + O(\Delta\overline{U}^2))(E_r^\top\overline{U})^{-1}.$$

We only assume $\mathbb{E}[\|\overline{X}\|_2^2] < \infty$ without assuming $\overline{X}$ is bounded. Thus, in order to take expectation over $\overline{X}$, we need a truncation argument. Write the SVD of $\overline{X}$ as $\overline{X} = V^\top SV$. Then $\overline{X}_n = V^\top(S \wedge n)V$ denotes the truncated $\overline{X}$ where $a \wedge b = \min(a, b)$ and $S \wedge n$ means to perform such an operation on each diagonal elements of $S$. Clearly, $\overline{X}_n$ has bounded norm $\|X_n\|_2 \le n$. Thus, we can take expectation with this truncated random variable $\overline{X}_n$. Moreover, as $n$ increases, $\|\overline{X}_n\|_2$ also monotone increases to $\|\overline{X}(t)\|_2$. Then by the monotone convergence theorem, $\lim_{n\to\infty} \mathbb{E}[\|\overline{X}_n\|_2^2] = \mathbb{E}[\|\overline{X}\|_2^2]$. This result allows us to take expectation on the infinitesimal increments $\Delta\widetilde{\gamma}_{i,s}^2(t)$.

Taking expectation conditioning on $\overline{U}_s$ and $z_{sh}$, then dividing both sides by $\eta$, we have

$$\mathbb{E}[\frac{\Delta\gamma_{i,s}^2}{\eta}|\overline{U}_s, z_{sh}] = 2e_i^\top\overline{U}(E_r^\top\overline{U})^{-1}(E_r^\top\overline{U})^{-\top}\overline{U}^\top(I + E)\Lambda e_i$$

$$- 2e_i^\top\overline{U}(E_r^\top\overline{U})^{-1}(E_r^\top\overline{U})^{-\top}(E_r^\top(I + E)\Lambda\overline{U})^\top(E_r^\top\overline{U})^{-\top}\overline{U}^\top e_i + O(\eta\|\Lambda\|_2^2)$$

$$= 2e_i^\top\overline{U}(E_r^\top\overline{U})^{-1}(E_r^\top\overline{U})^{-\top}\overline{U}^\top\Lambda e_i + 2e_i^\top\overline{U}(E_r^\top\overline{U})^{-1}(E_r^\top\overline{U})^{-\top}\overline{U}^\top E\Lambda e_i$$

$$- 2e_i^\top\overline{U}(E_r^\top\overline{U})^{-1}(E_r^\top\overline{U})^{-\top}(E_r^\top\Lambda\overline{U})^\top(E_r^\top\overline{U})^{-\top}\overline{U}^\top e_i$$

$$- 2e_i^\top\overline{U}(E_r^\top\overline{U})^{-1}(E_r^\top\overline{U})^{-\top}(E_r^\top E\Lambda\overline{U})^\top(E_r^\top\overline{U})^{-\top}\overline{U}^\top e_i + O(\eta\|\Lambda\|_2^2)$$

$$= 2\sigma_i\widetilde{\gamma}_i^2(t) - 2e_i^\top\overline{U}(E_r^\top\overline{U})^{-1}\Lambda_r(E_r^\top\overline{U})^{-\top}\overline{U}^\top e_i$$

$$+ 2e_i^\top\overline{U}(E_r^\top\overline{U})^{-1}(E_r^\top\overline{U})^{-\top}\overline{U}^\top E\Lambda e_i - 2e_i^\top\overline{U}(E_r^\top\overline{U})^{-1}(E_r^\top\overline{U})^{-\top}(E_r^\top E\Lambda\overline{U})^\top(E_r^\top\overline{U})^{-\top}\overline{U}^\top e_i$$

$$+ O(\eta\|\Lambda\|_2^2).$$

Under the geometric ergodicity condition, we know $\|E\|_2 \leq \tau$, which implies $\|E\Lambda\|_2 \leq \tau\sigma_1$. Then we have

$$e_i^\top \overline{U}(E_r^\top \overline{U})^{-1}\Lambda_r(E_r^\top \overline{U})^{-\top}\overline{U}^\top e_i \geq \lambda_r \widetilde{\gamma}_{i,s}^2,$$

$$\left|e_i^\top \overline{U}(E_r^\top \overline{U})^{-1}(E_r^\top \overline{U})^{-\top}\overline{U}^\top E\Lambda e_i\right| = O(\eta),$$

$$e_i^\top \overline{U}(E_r^\top \overline{U})^{-1}(E_r^\top \overline{U})^{-\top}(E_r^\top E\Lambda\overline{U})^\top(E_r^\top \overline{U})^{-\top}\overline{U}^\top e_i = O(\eta).$$

Combining the above three bounds, we have

$$\lim_{\eta\to 0}\mathbb{E}\left[\frac{\Delta\gamma_{i,s}^2}{\eta}\middle|\overline{U}_s, z_{sh}\right] \leq 2(\lambda_i - \lambda_r)\gamma_{i,s}^2.$$

This upper bound also implies that $\lim_{\eta\to 0}\mathbb{E}\left[\frac{[\Delta\gamma_{i,s}^2]^2}{\eta}\middle|\overline{U}_s, z_{sh}\right] = 0$, since the numerator is of order $O(\eta^2)$. Thus, we can show $\widetilde{\gamma}_{i,\eta}^2(t)$ converges weakly to the solution of

$$d\widetilde{\gamma}_i^2 = b_i\widetilde{\gamma}_i^2 dt \quad \text{with} \quad b_i \leq 2(\lambda_i - \lambda_r).$$

$\square$

### B.3 Proof of Theorem 3.5

*Proof.* We need the following lemma to bound the smallest eigenvalue of $\overline{U}^\top \Lambda \overline{U}$. Denote by $E_{\mathcal{A}_r} = [e_{a_1}, e_{a_2}, \ldots, e_{a_r}] \in \mathbb{R}^{m\times r}$ where $\mathcal{A}_r = \{a_1, a_2, \ldots, a_r\}$ denotes an index set of $\{1, 2, \ldots, m\}$ such that $a_1 > a_2 > \cdots > a_r$. Further denote by $\overline{\mathcal{A}}_r$ the complement of $\mathcal{A}_r$ in $\{1, 2, \ldots, m\}$ and write $\overline{E}_{\mathcal{A}_r} = E_{\overline{\mathcal{A}}_r}$. Additionally, write $\Lambda_{\mathcal{A}_r} = \text{diag}(\lambda_{a_1}, \lambda_{a_2}, \ldots, \lambda_{a_r})$ and $\overline{\Lambda}_{\mathcal{A}_r} = \Lambda_{\overline{\mathcal{A}}_r}$.

**Lemma B.1.** Suppose $\|\overline{E}_{\mathcal{A}_r}^\top \overline{U}\|_F^2 \leq O(\delta)$ with $\overline{U} \in \mathbb{R}^{m\times r}$ having orthonormal columns, then

$$\sigma_{\min}(\overline{U}^\top \Lambda \overline{U}) \geq \lambda_{a_r} - O(\delta)$$

*Proof.* Since $\|\overline{E}_{\mathcal{A}_r}^\top \overline{U}\|_F^2 \leq O(\delta)$, we have $\|E_{\mathcal{A}_r}^\top \overline{U}\|_2^2 = r - \|\overline{E}_{\mathcal{A}_r}^\top \overline{U}\|_F^2 \geq r - O(\delta)$. Therefore, $\|\overline{U}e_{a_i}'\|_2 \geq 1 - O(\delta)$ for $i \in \{1, \ldots, r\}$. Thus, we have

$$\begin{aligned}
e_j'^\top \overline{U}^\top \Lambda \overline{U}e_j' &= e_j'^\top \overline{U}^\top (E_{\mathcal{A}_r}^\top \Lambda_{\mathcal{A}_r}E_{\mathcal{A}_r} + \overline{E}_{\mathcal{A}_r}^\top \overline{\Lambda E}_{\mathcal{A}_r})\overline{U}e_j' \\
&\geq \lambda_{a_r}e_j'^\top \overline{U}^\top E_{\mathcal{A}_r}^\top E_{\mathcal{A}_r}\overline{U}e_j' + e_j'^\top \overline{U}^\top \overline{E}_{\mathcal{A}_r}^\top \Lambda\overline{E}_{\mathcal{A}_r}\overline{U}e_j' \\
&\geq \lambda_{a_r} - O(\delta).
\end{aligned}$$

$\square$

Now we turn to the proof of Theorem 3.5. We omit $(t)$ if there is no confusion. Denote by $\Delta\zeta_{ij,\eta}(t) = \zeta_{ij,\eta}(t+\eta) - \zeta_{ij,\eta}(t)$ and $\overline{Z}(t) = \eta^{-1/2}\overline{U}^\eta(t)$. We must show the mean and variance of $\Delta\zeta_{ij,\eta}(t)$ satisfies

$$K_{ij} = \lim_{\eta\to 0}\mathbb{E}\left[\frac{\Delta\zeta_{ij,\eta}(t)}{\eta}\middle|\overline{U}, z_{\lfloor t/\eta\rfloor h + h}\right] < \infty,$$

$$G_{ij}^2 = \lim_{\eta\to 0}\mathbb{E}\left[\frac{[\Delta\zeta_{ij,\eta}(t)]^2}{\eta}\middle|\overline{U}, z_{\lfloor t/\eta\rfloor h + h}\right] < \infty.$$

Then the sequence $\zeta_{ij,\eta}(t)$ weakly converges to the solution of the following SDE

$$d\zeta_{ij} = K_{ij}\zeta_{ij}dt + G_{ij}dB_t,$$

where $B_t$ is the standard Brownian motion. In fact, we have

$$\begin{aligned}
\mathbb{E}\left[\Delta\zeta_{ij,\eta}(t)|\overline{U}, z_{\lfloor t/\eta\rfloor h + h}\right] &= \mathbb{E}\left[\eta^{-1/2}e_j'^\top Q(\overline{Z}(t+\eta) - \overline{Z}(t))^\top e_i|\overline{U}, z_{\lfloor t/\eta\rfloor h + h}\right] \\
&= \eta^{1/2}e_j'^\top Q\mathbb{E}\left[\overline{U}^\top X - \overline{U}^\top XU\overline{U}^\top|\overline{U}, z_{\lfloor t/\eta\rfloor h + h}\right]e_i + O(\eta^{3/2}\|\Lambda\|_2^2) \\
&= \eta e_j'^\top Q\overline{Z}^\top(\Lambda + E\Lambda)e_i - \eta e_j'^\top Q(\overline{U}^\top(\Lambda + E\Lambda)\overline{U})\overline{Z}^\top e_i + O(\eta^{3/2}\|\Lambda\|_2^2) \\
&= \eta\lambda_i e_j'^\top Q\overline{Z}^\top e_i - \eta e_j'^\top Q(\overline{U}^\top \Lambda\overline{U})\overline{Z}^\top e_i + \eta e_j'^\top Q\overline{Z}^\top E\Lambda e_i - \eta e_j'^\top Q(\overline{U}^\top E\Lambda\overline{U})\overline{Z}^\top e_i \\
&\quad + O(\eta^{3/2}\|\Lambda\|_2^2) \\
&= \eta\lambda_i\zeta_{ij,\eta} - \eta e_j'^\top Q(\overline{U}^\top \Lambda\overline{U})\overline{Z}^\top e_i + \eta e_j'^\top Q\overline{Z}^\top E\Lambda e_i - \eta e_j'^\top Q(\overline{U}^\top E\Lambda\overline{U})\overline{Z}^\top e_i \\
&\quad + O(\eta^{3/2}\|\Lambda\|_2^2).
\end{aligned}$$

By the Lemma, we have $\sigma_{\min}\left(\overline{U}^\top \Lambda \overline{U}\right) \geq \lambda_{a_r} - O(\delta)$. Observe that $\|E\Lambda\|_2 = O(\eta)$, thus we obtain

$$\lim_{\eta \to 0} \mathbb{E}\left[\frac{\Delta \zeta_{ij,\eta}(t)}{\eta}\bigg| \overline{U}, z_{\lfloor t/\eta \rfloor h}\right] = K_{ij}\zeta_{ij,\eta} \text{ with } K_{ij} \in [\lambda_i - \lambda_1, \lambda_i - \lambda_{a_r}].$$

Note that when $j = r$, we have $K_{ir} = \lambda_i - \lambda_{a_r}$, because the equality $e_r'^\top Q(\overline{U}^\top \Lambda \overline{U})\overline{Z}^\top e_i = \lambda_{a_r} e_r'^\top Q\overline{Z}^\top e_i + O(\delta)$ holds. The variance is

$$\mathbb{E}\left[[\Delta\zeta_{ij,\eta}(t)]^2 | \overline{U}, z_{\lfloor t/\eta \rfloor h+h}\right] = \mathbb{E}\left[\left(\eta^{-1/2}e_j'^\top Q(\overline{Z}(t+\eta) - \overline{Z}(t))^\top e_i\right)^2 \bigg| \overline{U}, z_{\lfloor t/\eta \rfloor h+h}\right]$$

$$= \eta\mathbb{E}\left[\left(e_j'^\top Q\overline{U}^\top \overline{X}(I - \overline{U}\,\overline{U}^\top)e_i\right)^2 \bigg| \overline{U}, z_{\lfloor t/\eta \rfloor h+h}\right] + O(\eta^2\|\Lambda\|_2^2).$$

Observe that we have $\overline{U}^\top(I - \overline{U}\,\overline{U}^\top) = 0$, therefore, $\overline{U}Q^\top e_j'$ and $(I - \overline{U}\,\overline{U}^\top)e_i$ are orthogonal. Moreover, the norm of these two vectors satisfies $\left\|e_j'^\top Q\overline{U}\right\|_2 \approx 1$ and $\left\|(I - \overline{U}\,\overline{U}^\top)e_j'\right\|_2 \leq 1$. Hence, by the assumption that $\overline{X}$ has bounded second moment, we have

$$\lim_{\eta \to 0} \mathbb{E}\left[\frac{[\Delta\zeta_{ij,\eta}(t)]^2}{\eta}\bigg| \overline{U}, z_{\lfloor t/\eta \rfloor h+h}\right] < \infty.$$

$\square$

## B.4 Proof of Proposition 3.6

*Proof.* Since we start the algorithm at the saddle point and $K_{rr} = \lambda_r - \lambda_{r+1}$. The continuous time process $\zeta_{rr}(t)$ is approximately Gaussian distributed with mean 0 and variance $\frac{G_{rr}^2}{2K_{rr}}(e^{2K_{rr}t} - 1)$. We need the following condition,

$$\mathbb{P}\left(\left\|e_r^\top \overline{U}(t)\right\|_2^2 \geq \delta^2\right) \geq \mathbb{P}\left(\zeta_{rr}^2(t) \geq \eta^{-1}\delta^2\right),$$

which is equivalent to

$$\mathbb{P}(\zeta_{rr}^2(t) \geq \eta^{-1}\delta^2) = \mathbb{P}\left(\frac{|\zeta_{rr}(t)|}{\sqrt{\frac{G_{rr}^2}{2K_{rr}}(e^{2K_{rr}t} - 1)}} \geq \frac{\eta^{-1/2}\delta}{\sqrt{\frac{G_{rr}^2}{2K_{rr}}(e^{2K_{rr}t} - 1)}}\right).$$

Note that $\frac{\zeta_{rr,\eta}(t)}{\sqrt{\frac{G_{rr}^2}{2K_{rr}}(e^{2K_{rr}t}-1)}}$ converges weakly to $\frac{\zeta_{rr}(t)}{\sqrt{\frac{G_{rr}^2}{2K_{rr}}(e^{2K_{rr}t}-1)}}$, which is a standard Gaussian random variable. Let $\Phi(\cdot)$ denotes the standard Gaussian CDF, then we have

$$\eta^{-1/2}\delta \leq -\Phi^{-1}\left(\frac{1-\nu/2}{2}\right)\sqrt{\frac{G_{rr}^2}{2K_{rr}}(e^{2K_{rr}t} - 1)}.$$

Rearrange the above terms, we get

$$T_1 = \frac{1}{2(\lambda_r - \lambda_{r+1})}\log\left(\frac{2(\lambda_r - \lambda_{r+1})\eta^{-1}\delta^2}{[\Phi^{-1}(\frac{1-\nu/2}{2})]^2 G_{rr}^2} + 1\right).$$

$\square$

## B.5 Proof of Proposition 3.7

*Proof.* We know $\left\|\cos\Theta(\overline{E}_r, \overline{U}^\eta(t))\right\|_F^2 = \sum_{i=r+1}^m \gamma_{i,\eta}^2(t)$. Then using the upper bound $\widetilde{\gamma}_i^2(t)$, we have

$$\|\cos\Theta(\overline{E}_r, \overline{U}^\eta(t))\|_F^2 = \sum_{i=r+1}^m \gamma_{i,\eta}^2(t) \leq \sum_{i=r+1}^m \widetilde{\gamma}_{i,\eta}^2(t) = \sum_{i=r+1}^m \widetilde{\gamma}_i^2(0)e^{b_i t} \leq \sum_{i=r+1}^m \widetilde{\gamma}_i^2(0)e^{2(\lambda_r - \lambda_{r+1})t}.$$

In order for $\left\|\cos\Theta(\overline{E}_r,\overline{U}^\eta(T_2))\right\|_F^2 \leq \delta^2$, we need at most $T_2$ time such that

$$\sum_{i=r+1}^m \gamma_i^2(0)e^{2(\lambda_{r+1}-\lambda_r)T_2} \leq \delta^2.$$

Since the algorithm has escaped from the saddle point, we have $\left\|e_{r+1}^\top\overline{U}\right\|_2^2 \leq 1-\delta^2$ and $\left\|E_r^\top\overline{U}\right\|_2^2 \geq \delta^2$. Thus, the initial value satisfies $\sum_{i=r+1}^m \widetilde{\gamma}_i^2(0) \leq (1-\delta^2)\delta^{-2}$. Taking logarithm on both sides yields

$$T_2 = \frac{1}{2(\lambda_r-\lambda_{r+1})}\log\frac{1-\delta^2}{\delta^4} = \frac{1}{\lambda_r-\lambda_{r+1}}\log\frac{\sqrt{1-\delta^2}}{\delta^2}.$$

Then for a sufficiently small $\eta$, we have

$$\mathbb{P}\left(\sum_{i=r+1}^m \gamma_{i,\eta}^2(T_2) \leq \delta^2\right) \geq \frac{3}{4},$$

with $T_2 \asymp \frac{1}{\lambda_r-\lambda_{r+1}}\log\frac{1}{\delta^2}$. $\qquad\square$

## B.6  Proof of Theorem 3.8

*Proof.* The technique is almost the same as in Theorem 3.5. We have

$$\mathbb{E}\left[\Delta\zeta_{ij,\eta}(t)|\overline{U},z_{\lfloor t/\eta\rfloor h+h}\right] = \mathbb{E}\left[\eta^{-1/2}e_j'^\top Q(\overline{Z}(t+\eta)-\overline{Z}(t))^\top e_i|\overline{U},z_{\lfloor t/\eta\rfloor h+h}\right]$$

$$= \eta\sigma_i\zeta_{ij,\eta} - \eta e_j'^\top Q(\overline{U}^\top\Lambda\overline{U})\overline{Z}^\top e_i + \eta e_j'^\top Q\overline{Z}^\top E\Lambda e_i - \eta e_j'^\top Q(\overline{U}^\top E\Lambda\overline{U})\overline{Z}^\top e_i$$
$$+ O(\eta^{3/2}\|\Lambda\|_2^2),$$

and the variance satisfies

$$\mathbb{E}\left[(\Delta\zeta_{ij,\eta}(t))^2|\overline{U}(t),z_{\lfloor t/\eta\rfloor h+h}\right] = \mathbb{E}\left[\eta^{-1/2}e_j'^\top Q(\overline{Z}(t+\eta)-\overline{Z}(t))^\top e_i|\overline{U},z_{\lfloor t/\eta\rfloor h+h}\right]$$

$$= \eta\mathbb{E}\left[(e_j'^\top Q\overline{U}^\top\overline{X}(I-\overline{U}\overline{U}^\top)e_i)^2|\overline{U},z_{\lfloor s/\eta\rfloor h+h}\right] + O(\eta^2\|\Lambda\|_2^2).$$

Thus, with $\sigma_{\min}(\overline{U}^\top\Lambda\overline{U}) \geq \lambda_r - O(\delta)$ by Lemma B.1, we have

$$\lim_{\eta\to 0}\mathbb{E}\left[\frac{\Delta\zeta_{ij,\eta}(t)}{\eta}|\overline{U},z_{\lfloor t/\eta\rfloor h+h}\right] = K_{ij}\zeta_{ij,\eta}(t) \text{ with } K_{ij}\in[\lambda_i-\lambda_1,\lambda_i-\lambda_r],$$

$$\lim_{\eta\to 0}\mathbb{E}\left[\frac{[\Delta\zeta_{ij,\eta}(t)]^2}{\eta}|\overline{U},z_{\lfloor t/\eta\rfloor h+h}\right] < \infty.$$

$\qquad\square$

## B.7  Proof of Proposition 3.9

*Proof.* The proof is an application of Markov's inequality. Observe again that $\left\|\cos\Theta(\overline{E}_r,\overline{U}(t))\right\|_F^2 = \eta\sum_{i=r+1}^m\sum_{j=1}^r \zeta_{ij}^2(t)$. The expectation of $\zeta_{ij}^2(t)$ can be found as follows,

$$\mathbb{E}[\zeta_{ij}^2(t)] = \zeta_{ij}^2(0)e^{2K_{ij}t} + \frac{G_{ij}^2}{2K_{ij}}(e^{2K_{ij}t}-1)$$

$$\leq \zeta_{ij}^2(0)e^{2(\lambda_{r+1}-\lambda_r)t} + \frac{G_{ij}^2}{2(\lambda_r-\lambda_{r+1})}.$$

By Markov's inequality, we have

$$\mathbb{P}\left(\left\|\cos\Theta(\overline{E}_r,\overline{U}(t))\right\|_F^2 > \epsilon\right) \leq \frac{\mathbb{E}\left[\eta\sum_{i=r+1}^m\sum_{j=1}^r \zeta_{ij}^2(t)\right]}{\epsilon}$$

$$\leq \frac{\eta}{\epsilon}\sum_{i=r+1}^m\sum_{j=1}^r \zeta_{ij}^2(0)e^{2(\lambda_{r+1}-\lambda_r)t} + \frac{\eta}{\epsilon}r\frac{G_m}{2(\lambda_r-\lambda_{r+1})}.$$

Note that $\|\cos\Theta(\overline{E}_r, \overline{U}^\eta(t))\|_{\mathrm{F}}^2$ weakly converges to $\|\cos\Theta(\overline{E}_r, \overline{U}(t))\|_{\mathrm{F}}^2$, then we need at most $T_3$ time satisfying

$$\frac{\eta}{\epsilon}\sum_{i=r+1}^{m}\sum_{j=1}^{r}\zeta_{ij}^2(0)e^{2(\lambda_{r+1}-\lambda_r)T_3} + \frac{\eta}{\epsilon}r\frac{G_m}{2(\lambda_r-\lambda_{r+1})} \le \frac{1}{8}.$$

Rearrange and combine with $\eta\sum_{i=r+1}^{m}\sum_{j=1}^{r}\zeta_{ij}^2(0) \le \delta^2$, we get

$$T_3 = \frac{1}{2(\lambda_r-\lambda_{r+1})}\log\left(\frac{8(\lambda_r-\lambda_{r+1})\delta^2}{(\lambda_r-\lambda_{r+1})\epsilon - 4\eta rG_m}\right).$$

$\square$

## B.8 Proof of Corollary 3.10

*Proof.* We list the time upper bound given in the Stage 1, Stage 2 and Stage 3,

$$T_1 = \frac{1}{\lambda_r-\lambda_{r+1}}\log\left(\frac{2(\lambda_r-\lambda_{r+1})\eta^{-1}\delta^2}{[\Phi^{-1}(\frac{1-\nu/2}{2})]^2 G_{rr}^2} + 1\right),$$

$$T_2 = \frac{1}{2(\lambda_r-\lambda_{r+1})}\log\frac{1}{\delta},$$

$$T_3 = \frac{1}{2(\lambda_r-\lambda_{r+1})}\log\left(\frac{8(\lambda_r-\lambda_{r+1})\delta^2}{(\lambda_r-\lambda_{r+1})\epsilon - 4\eta rB}\right).$$

Choose the step size $\eta$ satisfies

$$\eta \asymp \frac{(\lambda_r-\lambda_{r+1})\epsilon}{5rG_m}.$$

With such a choice of $\eta$ and $\delta = O(\eta^{1/2})$, we have

$$\log\left(\frac{(\lambda_r-\lambda_{r+1})\delta^2}{(\lambda_r-\lambda_{r+1})\epsilon - 4\eta rG_m}\right) \asymp \log\frac{\lambda_r-\lambda_{r+1}}{rG_m}.$$

The total time $T$ is upper bounded by

$$T = T_1 + T_2 + T_3$$

$$= \frac{1}{2(\lambda_r-\lambda_{r+1})}\log\left(\frac{2(\lambda_r-\lambda_{r+1})\eta^{-1}\delta^2}{[\Phi^{-1}(\frac{1-\nu/2}{2})]^2 G_{rr}^2} + 1\right) + \frac{1}{2(\lambda_r-\lambda_{r+1})}\log\frac{1}{\delta}$$

$$+ \frac{1}{2(\lambda_r-\lambda_{r+1})}\log\frac{(\lambda_r-\lambda_{r+1})\epsilon\eta^{-1} - 4rG_m}{(\lambda_r-\lambda_{r+1})\delta^2}$$

$$\asymp \frac{1}{2(\lambda_r-\lambda_{r+1})} + \frac{1}{2(\lambda_r-\lambda_{r+1})}\log\frac{r}{\epsilon} + \frac{1}{2(\lambda_r-\lambda_{r+1})}\log\frac{1}{rG_m}$$

$$\asymp \frac{1}{\lambda_r-\lambda_{r+1}}\log\frac{rG_m}{\epsilon(\lambda_r-\lambda_{r+1})}.$$

$\square$