[Reviews · NeurIPS 2018]

Reviewer 1



This contribution concerns stochastic learning of covariance from time correlated samples. As such an important issues and the dependent sample issue is complex. The approach essentially is to subsample to avoid correlation and assume enough eigenvalue separation that the problem becomes "easy". Would have been more interesting to actually estimate both the time dependence and covariance in my view. Comment on author feedback: I have noted your promise to consider the issue in the next version and look forward to your future work addressing this obvious issue. Theory seems technically sound

Reviewer 2



Summary: The paper consider the setting of streaming PCA for time series data which contains two challenging ingredients: data stream dependence and a non-convex optimization manifold. The authors address this setting via downsampled version of Oja's algorithm. By closely inspecting the optimization manifold and using tools from the theory of stochastic differential equations, the authors provide a rather detailed analysis of the convergence behavior, along with confirming experiments on synthetic and real data. Evaluation: Streaming PCA is a fundamental setting in a topic which becomes increasingly important for the ML community, namely, time series analysis. Both data dependence and non-convex optimization are still at their anecdotal preliminary stage, and the algorithm and the analysis provided in the paper form an interesting contribution in this respect. The paper is well-structured, although I find some key definitions missing, and some technical parts somewhat hard-to-follow. Comments: L47 - I find 'Moreover, theses results..' somewhat vague. Can you further elaborate on the nature of the uncertainty not captured by current convergence analyses from your point of view? L115 - Many terms here are used without providing a definition first. E.g., $\mathcal{F}$ in $(\mathcal{S},\mathcal{f})$, positive recurrent, transition kernel,.. L162 - 'The eigengaps in ... is identfible' seems to have a grammatical mistake. L169 - Can you put your statements regarding the taxonomy of stationary points on a more rigorous ground? L274 - Most of the proof sketch addresses the continuous analog of the discrete dynamic, i.e., (stochastic) gradient flow. The derivation of the convergence rate for the original discrete setting is less clear to me. Also, I would suggest adding an explicit statement covering the conditions and convergence rates implied by the analysis, at earlier stages of the paper before the proof sketch (as well as other contributions made by the paper - such as the intimate analysis of the algorithm at different stage of the optimization process using SDE).

Reviewer 3



[Summary of the paper] This paper considers the problem of streaming PCA for stationary time series. The primary contributions of the paper are two-fold: first, the paper proposes to use down sampling to generate an estimator of the covariance matrix with small bias, addressing the problem of data dependency in time series setting; second, the paper provide a detailed characterization of the behavior of SGD on the streaming PCA problem, by approximate the dynamics of SDG using ODE and SDE. Numerical experiments that corroborate the theoretical analysis are also provided. [Quality] The paper appears to be technically correct (but I have not read through the appendix carefully). [Clarity] The paper is very well-written, and does a good job introducing relevant background. Some statements of the lemmas and theorems can be made more precise and complete. For example: -- in Lemma 2.2, the covariance matrix $\Sigma$ here is the covariance matrix of $z$ under the stationary distribution, which should be made clear. -- in Proposition 3.7, $T_2$ did not appear in the later statement. In the numerical experiment section, it would be nice if the paper can be more specific on how the step size is chosen. [Originality] Since I am not familiar with this field of study, I cannot accurately evaluate the originality of the paper. [Significance] In my opinion, the paper has made a valuable contribution, especially in the theoretical characterizations of SDG on streaming PCA problem using diffusion approximation, corroborated by numerical experiments. The analysis is challenging in that the problem itself is non-convex (although without any bad local minimas), and has degeneracy, which is also the case for many other problems in machine learning, and the way these challenges are tackled in this paper might shed some light on similar analysis of these problems.